# Therapy and Prevention for Human Toxocariasis

**DOI:** 10.3390/microorganisms10020241

**Published:** 2022-01-22

**Authors:** Jean-François Magnaval, Emilie Bouhsira, Judith Fillaux

**Affiliations:** 1Service de Parasitologie Médicale, Faculté de Médecine, Université de Toulouse, 37 Allées Jules-Guesde, 31000 Toulouse, France; 2Service de Parasitologie, Ecole Nationale Vétérinaire, 23 Chemin des Capelles, 31076 Toulouse, France; emilie.bouhsira@envt.fr; 3Service de Parasitologie-Mycologie, Hôpital Purpan, Centre Hospitalier Universitaire de Toulouse, 330 Avenue de Grande-Bretagne, 31059 Toulouse, France; fillaux.j@chu-toulouse.fr

**Keywords:** toxocariasis, treatment, benzimidazoles, albendazole, mebendazole, thiabendazole, diethylcarbamazine, prevention, emodepside

## Abstract

For the last four decades, knowledge about human toxocariasis with regard to its epidemiology, pathophysiology, clinical spectrum, and imaging or laboratory diagnosis has substantially progressed. Knowledge about specific therapy with anthelmintics has lagged behind. To date, only four drugs are registered for human use, and their efficacy has rarely been assessed in prospective controlled trials. It is likely that the repurposing of potent anthelmintics from veterinary medicine will improve this situation. Due to its wide availability and a lack of major side effects during short regimens, albendazole has become the drug of choice. However, its efficacy should be more precisely assessed. The role of anthelmintics in the treatment of neurological or ocular toxocariasis remains to be clarified. Prophylactic measures in humans or companion animals are efficient and represent first-line treatments for the control of this zoonosis. Unfortunately, their implementation in areas or countries where toxocariasis epidemiology is driven by poverty is quite difficult or unrealistic.

## 1. Introduction

Human toxocariasis is a worldwide zoonotic helminthiasis caused by infection with *Toxocara canis* or *Toxocara cati* larvae. Adult forms of these ascarid helminths parasitize canids or felids, [1] and they dwell in the upper digestive tract of these definitive hosts. Eggs passed in the feces must be in the soil for embryonation and subsequently become infective. Most frequently, humans become infected by ingesting embryonated eggs present in nearby soil [2] or on raw vegetables [3]. In addition to this telluric mode of contamination, direct transmission directly involving larvae through a foodborne route appears to be very likely. The literature reports some infections following the consumption of raw or undercooked meat, giblets or offal from potential paratenic homeotherm hosts such as chickens [4], cows [5], ducks [6], lambs [7], pigs [8], ostriches [9] or rabbits [8].

In the duodenum, larvae hatch from embryonated eggs or are released from animal tissues during the digestion process, penetrate the gut wall and migrate to the liver via the portal veins. They cross the liver and reach the lungs via pulmonary circulation. They re-enter systemic circulation, which transports them throughout the entire body, during which they can migrate for a variable period of time [2]. A large proportion of larvae enters a state of dormancy (hypobiosis) and can remain viable for several years [10]. Others become entrapped inside granulomas and subsequently destroyed by the host’s immunological response against soluble larval antigens of excretory–secretory origin (TES Ag) [11]. Whether dormant larvae are sensitive to anthelmintics available in human medicine remains unknown.

The infection of humans by *Toxocara* spp. larvae result in various syndromes. However, most cases likely are asymptomatic. According to the type of involvement, *Toxocara* infection can be classified as systemic (generalized) or compartmentalized. Systemic forms comprise major visceral *larva migrans* (VLM) syndrome and minor common/covert toxocariasis. Compartmentalized syndromes include ocular or neurological toxocariasis [12]. 

VLM was first described in 1952 in children presenting with an enlarged liver and elevated blood eosinophils [13]. The typical VLM patient is a child aged 2–7 years old with a history of geophagia and exposure to puppies in the home. The acute signs of VLM are associated with hepatic and pulmonary larval migration and mostly include abdominal pain, decreased appetite, restlessness, fever, coughing, wheezing and hepatomegaly. In this phase of the infection, there is usually a marked eosinophilia (>2000 cells/mm^3^), leukocytosis and hypergammaglobulinemia. In western countries, VLM syndrome is rarely encountered, and a review of the literature from 1952 to 1979 only found 970 reports [14]. 

Common or covert toxocariasis was identified in the middle of the 1980s by two case–control studies that were carried out in French adults [15] and Irish children [16]. In French patients, toxocariasis was clinically characterized by weakness, pruritus, rash, labored breathing and abdominal pain. Significant laboratory findings included eosinophilia (average 1444 cells per mm^3^), increased total serum IgE levels (average 851 IU/mL) and elevated antibody titers to *T. canis*. This syndrome was later termed “common toxocariasis” [17]. In Ireland, the most frequent clinical findings in children infected with *Toxocara* spp. larvae included fever, anorexia, headache, abdominal pain, nausea, vomiting, lethargy, sleep and behavior disorders, pharyngitis, pneumonia, cough, wheezing, limb pain, cervical adenitis and hepatomegaly. This disease syndrome in children was coined “covert toxocariasis” [16]. Common/covert toxocariasis has also been associated with various allergy-related syndromes including angioedema, chronic urticaria, pruriginous rashes and reactive arthritis [18]. 

Common/covert toxocariasis is mostly a benign, self-limiting infection that usually resolves within three months [19]. The resulting specific antibodies may persist for years, thus explaining the sometimes high seroprevalence rate. A recent meta-analysis of the literature showed that the rate ranged from 2% to 8% in urban areas of the Group of Seven (G7) countries. In rural settlements, the seroprevalence may reach 40%. In wet, tropical or subtropical countries, toxocariasis seropositivity has been found in up to 80% of the population [20].

Regarding the compartmentalized forms, ocular toxocariasis (OT) was first suspected by Wilder, who investigated nematode endophthalmitis in 1950 [21]. In five of Wilder’s cases, *T. canis* larvae were subsequently identified by Nichols [22]. OT typically occurs unilaterally in children and young adults. The most common signs or symptoms are visual loss, with onset over a period of days to weeks, leukocoria, vitritis, ocular injection and strabismus. Fundoscopy and biomicroscopic examination often disclose peripheral or central chorioretinal eosinophilic granulomas, other focal lesions on the posterior segment, or endophthalmitis [23]. In some individuals, these signs may come and go over a period of years. Certainly, the OT prevalence is underestimated, since the infection may be subclinical and only detected during a routine eye exam. In 2021, a meta-analysis suggested that the worldwide prevalence of OT was 9% when the results from various diagnostic techniques were combined and 1% when only an ophthalmic exam was performed [24]. 

Neurotoxocariasis (NT), the other compartmentalized form, is a rare disease. In 2016, a review of the literature found 100 cases of neurological involvement due to this zoonosis, as determined by the finding of *Toxocara* spp. larvae in cerebral spinal fluid (CSF), in brain tissue and the meninges and/or by immunodiagnosis using CSF [25]. The infection of the central nervous system (CNS) by *Toxocara* spp. larvae, when symptomatic, leads to major and sometimes letal events. Dementia, encephalitis, epilepsy, meningitis, myelitis or paresis/ tetraparesia have been reported [26,27].

*Toxocara* infection may elicit abnormalities in various laboratory tests [28]. Blood eosinophilia, although not specific to toxocariasis, has been constantly associated with VLM. In common/covert toxocariasis, blood eosinophilia may be lacking in some patients [17]. Moreover, this laboratory finding is often absent in patients with OT or NT.

A definitive laboratory diagnosis of human toxocaral infection uses the histopathological examination of tissue specimens including liver [29], lung, brain [26,27] and enucleated eye tissue [30]. A mobile larva or larval tract can be directly observed in the retina in patients with OT [31], and larvae were found in CSF [32]. Since histopathological examinations are mostly not feasible, the laboratory diagnosis of human toxocariasis relies upon serology. The most common immunodiagnostic test is the enzyme-linked immunosorbent assay (ELISA) with TES Ag from *T. canis* larvae maintained in vitro (TES-ELISA). However, some cross-reactivity has been reported in sera from multiparasitized patients in tropical regions [33]. A positive TES ELISA result should be confirmed by Western blot analysis (WB), which is as sensitive as ELISA, but more specific when bands from 24 to 35 kilodaltons are considered [28]. The use of recombinant TES Ag is thought to substantially increase the specificity of ELISA [34].

Both TES-ELISA and WB to detect specific anti-*Toxocara* IgG are unable to discriminate between current or past infections. This represents a major diagnostic pitfall, given the increasingly high levels of seroprevalence that have been reported worldwide. Consequently, circumstantial evidence of active systemic toxocariasis should be sought. Usually, the finding of both peripheral eosinophilia and a positive serologic test result is considered indicative of current infection. However, this implies that other causes of blood eosinophilia have been ruled out [35]. The diagnosis may be uncertain in individuals who lack eosinophilia but who present with one or more clinical signs of covert toxocariasis. In such cases, an increase in the concentration of serum total IgE > 500 IU/mL is further evidence of recent *Toxocara* infection [15] The detection of the eosinophil cationic protein (ECP), which is released by activated eosinophils, could be helpful [28].

Serum ELISA to detect anti-TES IgG for the diagnosis of OT is not fully informative due to two issues. First, a large proportion of patients who present with this compartmentalized zoonosis do not have detectable concentrations of specific IgG in serum. Second, because of the reported seroprevalence rate which is mostly related to the presence of residual specific antibodies, ocular involvement supporting another etiology may be found to be fortuitously associated with a positive result from *Toxocara* serology. Therefore, aqueous or vitreous fluid should be obtained when OT is suspected. The anti-TES Ag antibody concentration in these fluids has been found to be higher and of better diagnostic value than those in sera obtained from patients with OT [23,36]. NT is associated with a similar serology issue, and TES-ELISA should be performed on CSF.

## 2. Anthelmintics for Human Toxocariasis

Very few anthelmintics are available for the treatment of human helminthiases, and most of them have a veterinary origin. This situation is encountered in human toxocariasis, for which current therapeutic regimens only rely upon four licensed drugs.

### 2.1. Experimental Studies on the Sensitivity of Toxocara spp. Larvae to Anthelmintics

In vitro studies have focused on the assessment of the potential larvicidal activity of various compounds [37,38,39] and natural products [40,41,42]. The most interesting and recent findings involved some benzimidazole derivatives [43]. One compound showed higher nematocidal activity than albendazole (ABZ).

Many animal experiments to test the larvicidal effects of various anthelmintics on *Toxocara* spp. larvae were performed using rodent models, in which mice were orally infected with embryonated *Toxocara* spp. eggs [44,45,46,47,48,49,50,51,52,53,54,55,56]. The efficacy of these treatments relied upon larval recovery methods from artificially digested tissues of infected mice and untreated controls. Interestingly, only diethylcarbamazine (DEC) significantly reduced the survival rate of larvae [45,47]. Thiabendazole had a negligible larvicidal effect but produced the marked inhibition of larval migration through the tissues [49]. Other anthelmintics including ABZ, fenbendazole and levamisole caused significant larval retention in the liver, followed by the migration of very few larvae in the muscles and the brain of treated mice [50]. Most larvae retained in the liver subsequently died and were not recoverable. However, these findings do not have any direct relevance to human patients, for whom treatment is likely to begin long after the peak of hepatic and pulmonary larval migration.

An issue with the methodology hampers the informative value of these experiments. That is, mice weighing approximately 25 to 35 g were infected with large inoculums ranging from 500 to 2000 embryonated eggs. This corresponded to a dose of one million to 4 million eggs for a human adult. This situation would only be encountered among children with a history of geophagia or in patients with mental illnesses who lived or played in a highly contaminated environment. In contrast, the number of *Toxocara* spp. eggs required to produce common or covert toxocariasis in humans is likely much lower. For example, in 1959, a human volunteer was given approximately 100 *T. canis* embryonated eggs orally. His blood eosinophil count increased to 13,500 cells/ mm^3^ on day 30 post-infection and remained at 6150 cells/ mm^3^ at 4.5 months later. He also developed a persistent cough [57]. Moreover, to achieve a substantial reduction—over 90%—in mean tissue larval counts, the infected rodents were given a 30-day course of ABZ at 220 mg/kg body weight (b/w) daily, or fenbendazole at 750 mg/kg b/w daily or oxibendazole at 750 mg/kg b/w daily, respectively. In comparison, the usual dose for ABZ in humans is 10–15 mg/kg b/w daily. 

### 2.2. Efficacy of Available Anthelmintics for Therapy of Human Toxocariasis

The diagnosis of human toxocariasis was poor before the introduction of TES-ELISA in 1979 and remains challenging in many circumstances. Therefore, open prospective or retrospective studies and randomized trials are scarce. Only relevant studies including four or more cases are discussed below. Anecdotal reports are only quoted if they have a historical relevance or concern NT or OT.

The treatment of OT with anthelmintics represents a special situation. First, controversy concerning the safety of anthelmintic use for therapy of this compartmentalized form of toxocariasis has occurred for years and still persists. Some ophthalmologists have claimed that larval lysis will boost inflammation in the parasitized eye. Nonetheless, this pending issue has been answered by two experimental studies in mice [58] and in primates [59]. The results demonstrated that only viable *T. canis* larvae releasing TES Ag locally were responsible for ocular involvement. Whether anthelmintics would penetrate into the eye or permeate the blood–brain barrier is another question of crucial importance.

#### 2.2.1. Benzimidazole Derivatives

Benzimidazoles are a class of synthetic compounds with broad-spectrum anthelmintic activity. Four benzimidazole drugs can be used in humans: ABZ, mebendazole (MBZ), thiabendazole (TBZ) and triclabendazole, the latter being restricted to the treatment of fascioliasis [60]. Both ABZ and MBZ bind to free β-tubulin, which is the essential protein component of microtubules in worms. These drugs have a higher affinity for this component than for the human protein, thus leading to the inhibition of tubulin polymerization and the loss of cytoplasmic microtubules. Moreover, benzimidazoles disrupt glucose metabolism in some helminths [61]. TBZ targets NADH oxidase and fumarate reductase in worms [62].

##### Albendazole (ABZ)

ABZ (methyl1-5-propylthio-1 H-benzimidazol-2-ylcarbamate) was licensed in 1975 for use against several helminth species in domestic animals. Regarding use in humans, the first developmental studies found it to be active against most intestinal nematodes (*Ascaris lumbricoides*, hookworms and pinworms), so in 1982, ABZ was approved as a deworming medicine for human use. Subsequently, ABZ was licensed for the treatment of *Strongyloides stercoralis* infection and three helminthiases due to cestode species that infect humans during the larval stage (*Echinococcus granulosus* s.l., *E. multilocularis* and *Taenia solium*) [63]. ABZ was later recommended for the treatment of trichinellosis [64] and is currently employed as a macrofilaricide for the control of lymphatic filariases and onchocerciasis [65].

After oral ingestion, ABZ is poorly absorbed in the gastrointestinal tract and should be taken with fat [66]. ABZ is oxidized in the liver to its active metabolite, albendazole sulfoxide (ABZSO), for which blood levels are erratic. In healthy volunteers, it was demonstrated that these concentrations varied six-fold when assessed one hour after the ingestion of a 600 mg dose of ABZ. Variability still ranged from 1- to 4.5-fold at 24 h post-treatment [67]. This phenomenon may result in decreased efficacy and suboptimal cure rates.

The treatment of the generalized forms of toxocariasis with ABZ, without any concomitant use of corticosteroids, was the subject of a two-arm randomized trial. Nineteen Swiss patients in the ABZ arm were given 10 mg/kg b/w daily for 5 days [68]. Minor side effects, mostly mild asthenia and/or nausea, occurred in 37% of the patients. Clinical improvement was observed in 31.6% of the patients. Further studies were observational, prospective or retrospective. Sixteen Argentinian children presenting with VLM syndrome received ABZ at 15 mg/kg b/w daily for 15 days. Within two months, hepatomegaly along with imaging abnormalities on ultrasonography of the liver improved, and the level of hepatic enzymes normalized. The eosinophil count was 7947 ± 2164 cells/mm^3^ prior to treatment and decreased to 2349 ± 328 cells/ mm^3^ within one year (70.4% decrease). The authors considered a 100% efficacy to be achieved [69]. In Spain, four common/covert toxocariasis cases were diagnosed in immigrants from Latin America who received ABZ (10–15 mg/kg b/w for 5 days). The presence of side effects was not reported. Six months after the termination of therapy, the clinical picture had returned to normal in all patients. The mean eosinophil count was reduced by 52.7% [70]. A Polish trial included 66 children who were diagnosed with common/covert toxocariasis. They received ABZ at 15 mg/kg daily (maximum dose: 400 mg daily) in two courses lasting 5 days each, separated by a 10-day, treatment-free interval. Patients were checked 3 months later. The success rate for the cure of clinical signs or symptoms was 71% (47/66). Additionally, the authors claimed that the treatment reduced the mean eosinophil count and total IgE levels [71]. In Japan, a retrospective study analyzed 246 file records of toxocariasis patients, 113 of whom presented with VLM syndrome. The other patients had OT, NT or mixed type cases. Regardless of the form of toxocariasis, the ABZ regimen was 15 mg/kg b/w daily for 4 weeks, followed by a treatment-free interval of two weeks and then another 4-week course at the same daily dose. The overall rate of side effects, including the following major events, was 31%: hair loss (2 cases), liver dysfunction (32 cases) and skin rash (1 case). Minor adverse reactions were nausea (one case) or vomiting (one case). A follow-up study was carried out three to four months after the initial diagnosis. The authors’ main criterion for therapeutic success was the result of TES-ELISA serology, which had to be negative or show a 30% decrease in the serum anti-TES Ag antibody concentration. The presence of another criterion was required, namely the return of blood eosinophils to the normal level or the disappearance of clinical symptoms or abnormal results on medical imaging. Consequently, the authors claimed 80.5% efficacy of ABZ therapy in the VLM group [72]. Another observational study included 440 Korean patients with blood eosinophilia due to toxocariasis [19]. Only 153 presented clinical signs or symptoms. One hundred and forty-one symptomatic patients received 800 mg of ABZ daily, but the exact length of the regimen was not precisely defined (“≤1 week plus >1 week”). Twelve symptomatic patients were untreated. The clinical recovery rate was 85.1% (120 out of 141 patients) in those who were treated with ABZ and 83.3% (10 out of 12) in untreated patients (insignificant difference). Concerning blood eosinophilia, the improvement rates among treated and untreated symptomatic patients were not reported. Another Korean observational study analyzed the kinetics of blood eosinophilia in 34 toxocariasis patients treated with ABZ vs. 11 toxocariasis patients without treatment [73]. ABZ at 800 mg was given daily for 5 to 7 days (mean duration not available). Within 3 months, the eosinophil count reverted to normal values in 82.3% (28 out of 34) of patients in the ABZ arm vs. 36.4% (4 out of 11) of the untreated patients (Fisher’s exact test, *p* = 0.007).

These results suggested that the ABZ efficacy for the treatment of generalized forms of toxocariasis was clearly related to the length of therapy, provided that a daily dose of approximately 15 mg/kg b/w was administered. However, in the best scenario, ABZ treatment was unsuccessful in approximately one out of five patients [72]. This issue is likely rooted in the pharmacokinetics of the drug, which causes large variations in plasma levels of ABZSO among individuals [74]. 

Concerning OT therapy, a study on the pharmacokinetics of ABZ in patients treated for neurocysticercosis provided some circumstantial information about the possible penetration of ABZ and ABZSO in the eye [74]. A single 15 mg/kg dose of ABZ was administered to 18 patients with neurocysticercosis. The mean concentration of ABZSO was 0.918 ± 0.216 µg/mL (±standard error of the mean, SEM) in plasma and 0.392 ± 0.076 in CSF. The drug concentration in CSF compared to the plasma level was 43%, which made anti-parasitic activity likely.

For decades, the diagnosis of OT has remained challenging, which may explain the rarity of reports concerning anthelmintic therapy. Only one controlled, randomized study included OT cases in the patient panel [68]. Three patients were allotted to the ABZ arm. Unfortunately, when the authors reported the assessment of the efficacy of anthelmintic therapy, they did not stratify the cured patients according to generalized or ocular disease. In 2001, an Austrian retrospective study was carried out among five OT patients presenting with uveitis and retinochoroidal granulomas. They were given 800 mg of ABZ daily for two weeks combined with prednisolone, starting at 1.5 mg/kg, tapering over 3 months. No patient showed side effects attributable to the anthelmintic therapy. Post-treatment follow-up covered a mean period of 20.2 months. In all patients, the inflammatory process resolved and all the granulomas vanished [75]. In 2014, Korean authors published the results of a large retrospective cohort study [76]. They investigated the file records of 57 adult patients who had been diagnosed with OT on the basis of the presence of a retinal granuloma. Eleven patients did not receive any treatment. Other patients were treated according to three different drug regimens. Twelve patients were given ABZ at 800 mg daily for two weeks, twenty-three received ABZ plus corticosteroids and eleven were treated with corticosteroids alone. All were followed up for 6 months. The follow-up at three months suggested a trend toward better efficacy of the combination ABZ plus corticosteroids. An improvement in clinical symptoms was recorded in 14 out of 23 patients (60.9%) in this group vs. 4 out of 11 patients (36.4%) in the group that received corticosteroids alone and 3 out of 9 patients (33.3%) in the group that received ABZ alone. However, the distributions of the results were not significantly different (Fisher’s exact test).

Since NT is a rare disease [25], no controlled trials have been carried out. However, experience gained from the management of neurocysticercosis [77] suggests that ABZ in combination with corticosteroids would be effective for the treatment of NT. In particular, pharmacokinetics studies have demonstrated that ABZSO could be retrieved in CSF at efficient concentrations (see above). Moreover, the concomitant administration of corticosteroids (dexamethasone) with ABZ increased the plasma concentration of ABZSO by 50% [78]. When compared to other anthelmintics that also cross the blood–brain barrier, ABZ has an intestinal absorption rate 20-fold higher than that of MBZ [79]. ABZ is associated with a better tolerance than thiabendazole or diethylcarbamazine. Because of its pharmacological profile and its worldwide availability, ABZ was the most frequently used drug (67% of the cases) for treating NT. It resulted in an improvement in 47% of patients [25].

##### Mebendazole (MBZ)

MBZ (methyl-5-benzoyl-1H benzimidazol-2-ylcarbamate) is a broad-spectrum anthelmintic that was primarily synthesized and developed for veterinary use. In 1974, MBZ was approved by the Food and Drug Administration (FDA) for the treatment of *A. lumbricoides*, *Enterobius vermicularis*, hookworm species and *Trichuris trichiura*. In 1975, experimental works in mice and prospective studies in humans demonstrated that MBZ was active against *Echinococcus granulosus* protoscoleces, thus paving the way for the medical treatment of hydatidosis [80].

MBZ is practically insoluble in water and should be administered with a fatty meal which increases the plasma levels by 5-fold. The drug undergoes extensive first pass metabolism in the intestinal wall and liver. Metabolites found in plasma lack any anthelmintic activity. Similar to ABZ, the plasma levels showed marked interindividual variations. The coadministration of cimetidine and MBZ in healthy adults was found to increase the plasma concentrations of MBZ. This phenomenon was possibly due to the inhibition of hepatic first pass cytochrome P450-mediated metabolism of the anthelmintic [81]. However, the adjunction of cimetidine to MBZ did not result in an increase in plasma levels of MBZ or its metabolites in 17 patients with hydatidosis [82] and 12 patients with toxocariasis [83].

Various MBZ regimens for treating common/covert toxocariasis in adults have been tested in three randomized trials. In France, MBZ was given at 25 mg/kg b/w daily for 7 days [84] or 20–25 mg/kg b/w daily for 3 weeks [85]. A discontinuous regimen, namely 10-15 mg/kg b/w daily for 3 consecutive days in a week for 6 weeks, was tested against a placebo to assess MBZ efficacy on dormant *Toxocara* spp. larvae [86]. In this trial, the efficacy of MBZ on the clinical score was found to be similar to that of the placebo, but the laboratory score was significantly reduced by 17.7%, whereas an increase was observed in the placebo arm. Conversely, continuous regimens with a higher daily dose resulted in a greater reduction in the intensity of clinical manifestations and mean blood eosinophil count [84,85]. Minor side effects consisting of weakness, dizziness, nausea and abdominal and gastric pain were mild and occurred in 17% to 58% of patients. 

The use of MBZ for OT or NT is anecdotal and was only reported once for each form.

##### Thiabendazole (TBZ)

TBZ, or tiabendazole (2-thiazol-4yl-LH benzimidazole), was originally introduced in 1961 in veterinary medicine for the treatment of many roundworm infections in livestock and poultry. In 1963, TBZ was recognized as the first effective drug for the treatment of strongyloidiasis in humans, and ascariasis and enterobiasis were successfully targeted [87,88]. Anti-fungal properties of TBZ were reported in 1964 [89]. Additionally, experimental studies in noninfected rats demonstrated that TBZ had anti-inflammatory, anti-pyretic and analgesic properties. These findings subsequently correlated with the results of clinical observations among patients treated with TBZ for various helminthiases [90]. 

In 1967, TBZ was approved for human use in the United States. For the treatment of intestinal helminth infections, the recommended regimen was 50 mg/kg b/w daily for two days, and the maximum daily dose was 3 g [88]. Moreover, in 1969, TBZ was registered in the United States as a pesticide and was intensively used worldwide as an agricultural fungicide for the post-harvest treatment of fruits and vegetables [91].

TBZ has a low solubility in water but is readily soluble in diluted acid and alkali. Following the ingestion of 1 g of TBZ, the drug is quickly absorbed then metabolized in inactive metabolites which account for 97% of the plasma concentrations. Within 24 h, 80% of intact TBZ and its metabolites are excreted in urine [92]. 

The efficacy of TBZ for treating human toxocariasis was assessed in two two-arm randomized trials. When tested vs. MBZ at 25 mg/kg b/w daily for 7 days, TBZ therapy elicited a clinical improvement in 50% of 30 French patients [84]. The level of blood eosinophilia was not affected. The presence and nature of side effects were not discussed.

In a Swiss trial, TBZ was given at 50 mg/kg b/w daily for 7 days to 15 patients. Assessed on the clinical signs or symptoms, the success rate was 53.3%. Minor side effects (anorexia, nausea and dizziness) were observed in 4 out of 15 patients (27%) [68]. In a more recent Vietnamese study, TBZ was administered at a dosage of 50 mg/kg b/w daily for 7 days. In total, 79% of the 80 patients achieved cure, and 33.8% reported known side effects (see above) which were mostly neurological [93].

Animal experiments showed that TBZ penetrated rabbit eye tissues after instillation in the eye sac or via an ophthalmic ointment [94]. In 1990, an OT patient was given TBZ orally at a single dosage of 25 mg/kg 90 min prior to surgery. The detection of the drug in aqueous fluid and vitreous humor found that concentrations of TBZ in the vitreous were within the range of in vitro anti-parasitic activity for most helminths [95]. However, the role of TBZ in chemotherapy for OT remains undefined since no prospective trials have been carried out. A retrospective study covering 20 years of management of pediatric OT in Costa Rica analyzed five cases that were treated with TBZ. No improvement of the best corrected visual acuity was observed [96]. 

Pharmacological investigations were carried out in a patient with disseminated strongyloidiasis (hyperinfection syndrome) who had received several courses of TBZ. They demonstrated that TBZ and metabolites crossed the blood–brain barrier [97]. However, TBZ, sometimes in combination with corticosteroids, has been sparsely employed for the treatment of NT, and the results have been mixed (review by Moreira [98]).

TBZ is no longer approved for human use in the United States and in most countries in the European Union.

#### 2.2.2. Diethylcarbamazinze (DEC)

DEC (diethyl-4-methylpiperazine-1-carboxamide) is a synthetic piperazine derivative that was marketed in 1947 as an anti-filarial agent and has remained the mainstay for the treatment of various filariases for decades. DEC is water soluble and readily absorbed from the small intestine following oral administration. Plasma concentrations peak within 1–2 h, and the half-life is attained within 10–12 h. Its metabolites do not have any anti-filarial effect. Experiments in mice showed that DEC permeated the blood–brain barrier and accumulated quickly in the brain. When DEC solutions were applied locally to the eyes of rabbits, the compound entered the aqueous humor and achieved high concentrations [99,100].

The mechanism of action of DEC is complex and remains partially unclear. DEC is recognized as an anti-inflammatory and immunomodulatory agent [101]. Regarding specific anti-parasite activity, DEC enhances both the adherence and cytotoxicity of neutrophils and eosinophils to microfilariae by altering the parasite’s surface layer. This action requires the presence of specific antibodies [102]. Additionally, DEC activates platelets that release free radicals. This action is antibody-independent and is triggered by a filarial excretory antigen [103]. A direct anti-helminthic effect has also been demonstrated in vitro on *Wuchereria bancrofti* microfilariae. It was characterized by morphological alterations such as loss of the microfilarial sheath and lysis of the cytoplasm together with the destruction of organelles and the formation of vacuoles [104].

Therapy of VLM with DEC was advocated as soon as 1964 [105], but no further reports were issued during the rest of the decade. In 1971, the results of DEC therapy for three pediatrics (<5 y-old) patients with *Toxocara* infection were published. Two young patients presented with typical VLM signs and symptoms, and one only exhibited asymptomatic blood eosinophilia. According to the classical regimen used for lymphatic filariasis, DEC was given in gradually increasing doses up to 150 mg daily for three weeks. No side effects were reported. Follow-up clinical examinations and blood count results suggested that VLM cases were cured, but the eosinophilia level in the asymptomatic child was not affected [47]. 

In a two-arm, randomized study, the efficacy of DEC for the treatment of common/covert toxocariasis was compared to that of MBZ. The DEC regimen was 3–4 mg/kg b/w daily for 21 days. The therapeutic schedule started at 25 mg daily, and the dose was progressively increased to avoid adverse reactions due to larval lysis. No antihistamine drugs were used. Patients were followed between one and two months after the end of therapy. On average, this regimen resulted in a 70.9% significant decrease in the intensity of clinical signs, and mean blood eosinophilia was significantly reduced by 40.9%. Thirty-one percent of the patients complained of minor side effects, namely increased weakness, dizziness, nausea, or vomiting. These effects were dose-dependent and waned when the daily dose was tapered. In 4 out of 39 patients, a Mazzotti-like reaction (itching, urticaria and/or edema) was observed, suggestive of accelerated larval lysis. One patient experienced severe gastric pain, and treatment was terminated [85].

DEC for OT has been mostly reported as single case reports. Unfortunately, great variations in therapeutic associations and durations of treatment have made it impossible to assess the impact of DEC on the clinical outcome. Only two open studies were homogeneous regarding the regimen used. Among a group of 30 Polish children who received DEC (6 mg/kg b/w daily for 21 days) plus corticosteroids, improvements in visual acuity were achieved in 12 patients (40%) [106]. Another observational trial was conducted in 36 adult patients in Japan. DEC was combined with prednisolone and administered at 300 mg daily for one week followed by 600 mg daily for 7 weeks. Ocular inflammation receded in all 36 patients, but this positive general outcome was attributed by the authors to the corticosteroids rather than to DEC [107].

Regarding NT, the diversity of treatment schedules that are reported in the literature hinders assessment of the efficacy of DEC. Review articles suggest that DEC use is possible [25,26,27].

#### 2.2.3. Ivermectin

Ivermectin is a synthetic compound and a derivative of avermectin B1, a macrocyclic lactone that was developed from the cultivation of the bacterium *Streptomyces avermitilis*. Ivermectin is a broad-spectrum anthelmintic, but it is also active against parasitic arthropods, insects and mites. In 1981, ivermectin was marketed for veterinary medicine, and then, the drug maker issued compassionate use in humans, namely providing ivermectin freely to programs dedicated to the control of onchocerciasis. Ivermectin revolutionized the therapy of various human filariases. Subsequently, the drug was licensed in the European Union and the United States for the treatment of strongyloidiasis, and in the European Union for the treatment of scabies.

When ivermectin is administered orally, it strongly associates with digesta, which reduces the potential for absorption and elicits a great variability in the half-life [108]. Macrocyclic lactones do not cross the blood–brain barrier, as they are excluded by a P-glycoprotein drug pump [109]. Ivermectin and other macrocyclic lactones exert anthelmintic action by opening glutamate-gated chloride channels in invertebrates [110].

Because ivermectin may be given in a single 12 mg dose and has few side effects, the drug was sporadically used for the treatment of human toxocariasis. When tested in an open prospective trial including 15 French cases of common toxocariasis, ivermectin was only 40% effective in reducing clinical manifestations, and there was no significant decrease in the blood eosinophil count [111]. In another observational study [72], seven VLM patients were treated with ivermectin. Three were cured (70%), but the authors did not provide any detailed information about this trial. Thus, ivermectin should not be used for the treatment of human toxocariasis, particularly OT, until the question of its efficacy has been evaluated in controlled trials.

#### 2.2.4. Levamisole

Levamisole is the levorotatory enantiomer of tetramisole (I-tétrahydro-2,3,5,6 phényl-6 imidazol [2,1–6] thiazole), a synthetic imidazothiazole that was first used in the early 1960s as an anthelmintic agent in veterinary medicine. In humans, the anthelmintic efficacy of levamisole against *A. lumbricoides* and hookworms was first reported in 1970 [112]. Then, levamisole was found to have significant activity against a larger array of helminths, including the microfilariae of *W. bancrofti* and *Brugia malayi* [113]. Since levamisole has immunomodulatory effects, the drug has been assessed for the treatment of various immune-mediated diseases. Levamisole has also been used in combination with other compounds for the treatment of various malignancies, with equivocal results. However, the occurrence of severe adverse reactions, such as agranulocytosis or bone marrow aplasia, discredited the drug [114].

Levamisole has rarely been employed for the treatment of human toxocariasis, with its use only recorded in one case of common/covert toxocariasis and one case of OT.

In 2000, levamisole was withdrawn from the American market and is no longer available in most countries in the European Union. 

#### 2.2.5. Nitazoxanide

Nitazoxanide (2-acetyloxy-N-(5-nitro-2-thiazolyl) benzamide) has broad anti-parasitic efficacy and is licensed in the United States for the treatment of *Cryptosporidium* spp. or *Giardia* spp. infections. In clinical trials, the drug has been effective against a broad array of helminthiases [115]. Concerning toxocariasis specifically, nitazoxanide induced a significantly lesser reduction in larval load in mice infected with *T. canis* larvae than MBZ [116]. The efficacy of nitazoxanide for the treatment of human toxocariasis remains to be assessed.

Table 1 summarizes the data from the main above-cited clinical studies or trials.

#### 2.2.6. Future of Anthelmintic Therapy for Human Toxocariasis

Human toxocariasis is a worldwide neglected zoonosis that is globally related to poverty [117]. Lower income, malnutrition sometimes inducing geophagia, poor environmental or personal hygiene, lack of clean drinking water or a sewage system and the presence of roaming cats and dogs in the local environment are recognized as major risk factors for this helminthiasis. A wet and warm climate worsens the situation. For people living in such areas, the control of toxocariasis will not be achieved by novel anthelmintics but by a drastic improvement in general economics and personal statuses. In industrialized countries of the G7, toxocariasis is neglected since this zoonosis mostly assumes the features of a benign disease, and severe forms such as NT, OT or VLM are rarely diagnosed. Toxocariasis is even coined as an “orphan disease” by Orphanet, a European website providing information about rare diseases [118]. 

Given this situation, any progress in the armamentarium against human toxocariasis solely relies upon the improvement of existing drugs or from the repurposing of broad spectrum anthelmintics [119] used in veterinary medicine. Fortunately, momentum for both approaches has been increased by large-scale programs for the control of soil-transmitted helminthiases (STHs) or human filariases. 

New formulations of ABZ aim to improve the intestinal absorption and bioavailabilty of the drug. For example, ABZ was included in chitosan microspheres and then orally administered to mice infected with embryonated *T. canis* eggs. The result was a significant reduction in the number of migrating larvae [56]. The inclusion of ABZ in orally administered liposomes blocked the development of *E. granulosus* in mice. Conversely, infection progressed in animals treated with a classical ABZ suspension [120]. Moreover, such liposomal formulations increased the brain concentration of ABZSO, the active metabolite of ABZ, in healthy mice [121]. Obviously, infection caused by larval cestodes is the designated target of such new formulations of ABZ, but other indications such as human toxocariasis may be considered. 

Though unavailable to date in human medicine, potent benzimidazole compounds are routinely employed by veterinarians for the deworming of livestock. ABZSO is marketed as ricobendazole (RBZ). In combination with ivermectin, in cattle, ABZSO eliminated nematode worms that were resistant when only one of the anthelmintics was administered [122]. Oxfendazole (OFZ) is the sulfoxide metabolite of fenbendazole. OFZ is a broad-spectrum anthelmintic that is currently marketed for use against parasitic nematodes in ruminants’ respiratory and digestive tracts. Experimental studies demonstrated OFZ activity against *T. solium* cysticerci in pigs [123] or adult filarial worms in rodents [124]. In healthy human volunteers, OFZ showed fast intestinal absorption followed by a long half-life in plasma. Regardless of the administered dose, neither clinical side effects nor adverse impacts on the liver, kidney or bone marrow were recorded [79]. These characteristics make OFZ a candidate for the transition from veterinary medicine to human use. Through the Drug for Neglected Diseases Initiative and the Helminth Elimination Platform [125], OFZ is currently being tested as a macrofilaricidal treatment for onchocerciasis. 

Milbemycins are macrocyclic lactones that were isolated in 1967 from fermentation of the soil bacterium *Streptomyces hygroscopes*. Moxidectin (MOX) is a chemical derivative of nemadectin, a member of the milbemycin family that is produced by *Streptomyces cyaneogriseus*. Although the range of targeted infectious agents is rather similar to the ivermectin spectrum, the half-life of MOX is longer [126]. In 2018, MOX received approval in the United States for the treatment of onchocerciasis. 

Repurposing of existing anthelmintics is usually prompted by assessments in various control programs. This methodology has made compounds from the veterinary world available for human medicine. The suitability of these ‘novel’ anthelmintics for toxocariasis therapy can only be determined by prospective controlled studies. Meanwhile, tentative individual treatments that would use the aforementioned compounds should be avoided, particularly in patients with compartmentalized forms of toxocariasis.

## 3. Adjunctive Therapies

Serious or fatal complications in people caused by the systemic migration of *Toxocara* spp. larvae, including heart failure due to, e.g., endomyocardial fibrosis, myocarditis, pericarditis and cardiac tamponade or lung involvement with respiratory obstructive syndrome have been reported. The neurological migration of larvae may also be life-threatening. Patients presenting with such syndromes fall primarily in the scope of intensive care medicine, and their management will not be discussed further. Similarly, surgical treatments for OT are the choice of ophthalmologists in charge of the patients. 

Although no controlled trials of the single use of corticosteroids have been carried out, empirical and generalized agreement exists for the use of these adjunctive drugs in human toxocariasis, particularly for OT or NT patients. Intermediate-acting glucocorticosteroids (prednisone or prednisolone) with lower anti-inflammatory activity [127] are indicated for long-term treatments of OT. Long-acting compounds with 30-fold greater anti-inflammatory activity are preferentially used for NT. Triamcinolone is usually chosen for subconjunctival instillations.

## 4. Indications for Anthelmintic Therapy

Whether a patient with systemic toxocariasis needs anthelmintic therapy depends on the syndrome. All children and adults with VLM should be treated. Patients presenting with common/covert toxocariasis who have blood eosinophilia do not necessarily require a specific treatment. This form of toxocariasis is mostly self-limiting within 3 months [19] provided appropriate preventive measures ruling out identified risk factors are implemented. Asymptomatic toxocariasis cases with blood eosinophilia only require thorough prevention. Patients exhibiting a clinical pattern consistent with common/covert toxocariasis but without blood eosinophilia represent a diagnostic dilemma. ECP is a marker indicative of a mass of activated eosinophil cells in the whole body, not only in blood. The detection of ECP concentrations could therefore provide key information about this situation [28,128]. Specifically, the avidity index of anti-TES IgG antibodies is claimed to discriminate between recent and past infections [129,130]. However, the reliability of the information supplied by this assay should be further assessed.

The role of anthelmintic treatment in compartmentalized toxocariasis, OT and NT, remains to be clarified. Regarding OT, experimental studies have demonstrated the harmful role of viable *Toxocara* spp. larvae, thus suggesting that drug-induced parasite death would reduce local inflammation [58,59]. Unfortunately, the great majority of published small series or anecdotal reports about anthelmintic therapy for OT referred to the concomitant use of anthelmintics and corticosteroids. The same consideration applies to NT. Review articles about NT and OT did not answer this question. Consequently, a meta-analysis investigating the efficacy of reported drug regimens (anthelmintics alone, corticosteroids alone, or both) for NT and OT is urgently needed.

## 5. Drug of Choice, Treatment Regimens and Therapeutic Monitoring

The few prospective controlled or observational trials that have been performed to evaluate therapy for toxocariasis have occasionally reported inconsistent dosages and suboptimal durations. The provided information is therefore somewhat questionable. Nonetheless, ABZ appears to be the drug of choice due to its quasi-worldwide availability, low price, seemingly acceptable efficacy and lack of major adverse reactions. We recommend 10–15 mg/kg b/w daily for 14 days. A higher daily dose and an excessively prolonged course might cause severe and sometimes irreversible adverse reactions (agranulocytosis, aplasia or hepatotoxicity), such as those reported in the treatment of echinococcoses with benzimidazoles [131], and nonadherence to therapy. However, the heterogeneity of studies on ABZ makes our recommendation open to discussion, and the optimal dose and duration of the drug course remains to be established. 

ABZ failure in a given patient might be due to a lack of absorption of benzimidazole derivatives (see above), so MBZ would not be a better option. DEC could be used as an alternative agent at a dose of 4–6 mg/kg daily for 21 days. 

The evaluation of treatment efficacy in toxocariasis patients primarily relies upon the clinical response. Consequently, the point during the course of the infection at which the treatment is initiated as well as the frequency of follow-up examinations are critical and depend on the type of toxocariasis. For example, when the efficacy of DEC for the treatment of common/covert toxocariasis was evaluated [85], a significant clinical improvement was noticed when physical examination was performed at approximately one month of follow-up. In patients presenting with a more severe clinical picture and treated with TBZ, the optimal point for follow-up was 3 months. Longer follow-up durations only induced a marginal increase in the success rate [93]. Moreover, reinfections might occur if preventive measures are not fully implemented by patients, thus hindering the assessment of anthelmintic efficacy [71]. The use of a quantitative clinical scoring system, as described in randomized studies [85,86], is helpful for statistical evaluations of the evolution of the clinical picture. Abnormal patterns that can be detected on imaging techniques in organs or tissues [132] may be considered along with clinical signs in the assessment of treatment efficacy. However, great variability exists in the post-treatment evolution of these imaging abnormalities.

Among nonspecific laboratory tests, the eosinophil count appears to be the most helpful for post-treatment follow-up of generalized toxocariasis, according to the above-cited prospective trials and observational studies. As reported for other tissue-dwelling helminthiases, a rapid increase in the level of blood eosinophilia can be observed within one week following the termination of anthelmintic therapy. This phenomenon is attributed to the lysis of the worms in their adult or larval stages and is significantly related to the parasitic load [133]. When present, it resolved within 1 month in a series of toxocariasis patients [93]. An elevated level of serum total IgE over 500 IU/mL is recognized as a valuable indicator of the presence of tissue-dwelling helminths, including *Toxocara* spp. larvae [28,35]. Because post-treatment kinetics of IgE are far slower than that of eosinophil count, the mean serum concentration of this class of immunoglobulin remained unchanged within 1 month after anthelmintic treatment [85] but regressed significantly within 3 months [93]. Therefore, assessment of the concentration of serum total IgE should be performed during post-treatment follow-up. 

The usefulness of TES-ELISA results for monitoring the success of anthelmintic therapy is questionable. In two case series including children with toxocariasis treated with either TBZ (11 subjects) or no treatment (9), a 1-year period was required to observe a significant decrease in the mean concentration of specific IgG [134]. Further studies provided controversial results: 91 out of 113 Japanese adult patients showed a 30% reduction in ELISA optical densities within 4 months, but in 26 Brazilian children, serology by ELISA normalized after 4 years [135]. Although these results are heterogeneous, a trend arises that suggests ELISA for the detection of anti-TES IgG is not suitable for post-treatment follow-up due to generally very slow kinetics. WB for the detection of specific IgG is a quantitative assay that delivers categorical results. The number of bands observed in a given pattern does not correlate with the serum load of specific antibodies but to the variety of immunological specificities of these immunoglobulins. Fifteen Brazilian toxocariasis patients underwent a serological follow-up with WB for up to 116 months following anthelmintic therapy. On average, the WB result turned negative within 52.1 ± 21.3 months [136]. WB using TES Ag does not seem appropriate for monitoring post-treatment follow-up. 

## 6. Prevention of Toxocariasis

### 6.1. In Humans

To date, the main epidemiological factors modulating the transmission of human toxocariasis have been identified [1]. Consequently, individual preventive measures have proven efficient. Regardless of the clinical form of toxocariasis or the chemotherapy regimen used, measures should be taken to prevent reinfection. The patient or their surrogates should be questioned carefully to identify personal risk factors for *Toxocara* infection and to find likely sources of *Toxocara* spp. eggs in the environment. Risk factors for infection include behaviors such as geophagia and poor personal hygiene. Dogs or cats in the patient’s environment should be dewormed periodically (see Section 6.2.). Pet owners dwelling in suburbs or rural areas should be informed that their animal companions are at a greater risk of becoming infected with *Toxocara* spp. [137]. In such areas, dogs and cats prey frequently on small rodents or birds that can act as paratenic hosts in *Toxocara* spp. transmission. The ingestion of tissues possibly containing *Toxocara* spp. larvae may result in repeated infections in definitive canine or feline hosts. Moreover, any contaminated soil should be removed or the area should be closed so it is not accessible to small children.

Household gardens should be fenced to eliminate contamination by dogs. If roaming cats are present in the vicinity, areas for growing lettuce should be covered by appropriate materials as soon as the soil is turned for seeding. Cats are indeed attracted by soft soils for defecation. Sand boxes also should be protected. Vegetables or fruits gathered in possibly contaminated gardens should be thoroughly washed before eating [138]. Raw or undercooked meat, giblets or offal from paratenic hosts that could harbor *Toxocara* spp. larvae should be avoided [139]. Remedies from traditional or alternative medicines that are based upon the ingestion of raw wild invertebrates should be prohibited [140]. Parents and children should receive counseling for geophagia. Personal hygiene, including handwashing, is important, especially when handling foods and after contacting dogs. While gardening, wearing appropriate gloves is of paramount importance.

### 6.2. In Definitive Canine or Felid Host

Anthelmintics that are active against adult *Toxocara* worms are classified into four groups: benzimidazoles/probenzimidazoles, cyclooctadepsipeptide, macrocyclic lactones and tetrahydropyrimidine/imidazothiazoles. Emodepside was more recently identified and is a member of the cyclooctadepsipeptide class.

Only compounds that can diffuse into tissues should be used to eliminate migrating larvae. Among this group, which includes benzimidazoles (albendazole, fenbendazole and oxfendazole), levamisole, emodepside and one macrocyclic lactone (selamectin), fenbendazole likely has the greatest efficacy against migrating *T. canis* larvae in dogs. Administered to greyhounds at 50 mg/kg b/w daily for three days, fenbendazole reduced the numbers of third- and fourth-stage *T. canis* larvae by 94.0% (52]. Emodepside is the best option for the treatment of migrating *T. cati* larvae in cats. Assessed in an international multicenter trial, emodepside in topical solution was 96.8% effective against third-stage larvae and at least 99.4% effective against fourth-stage larvae, respectively [141]. In pregnant queens, a topical formulation of emodepside should be administered before parturition to prevent lactogenic transmission of *T. cati* to kittens [142].

Prophylactic deworming should always be performed according to the advice of a veterinary practitioner because the choice of the anthelmintic drug and the therapeutic schedule vary according to animal age and status. In Europe, detailed and constantly updated guidelines to assist both veterinarians and pet owners are available from the European Scientific Counsel for Companion Animal Parasites (ESCCAP). ESCCAP is a consortium of internationally renowned experts who specialize in the control of parasite infections in dogs and cats and provides research-based, independent advice [143].

## 7. Conclusions

To date, anthelmintic therapy for human toxocariasis remains rather unsatisfactory. There are few available compounds, and they are out of date. Their efficacy has rarely been assessed in prospective controlled trials. Clearly, anthelmintic treatment of the systemic forms of human toxocariasis should be standardized. A scoring system to quantify clinical severity should be developed by expert consensus so that therapeutic efficacy may be precisely assessed.

The role of anthelmintics in the treatment of NT or OT should be clarified. Because both forms are not frequently diagnosed, prospective studies are not truly conceivable. However, a meta-analysis of the published works on anthelmintic therapy for NT or OT would improve our knowledge of this topic.

## Figures and Tables

**Table 1 microorganisms-10-00241-t001:** Characteristics and results of studies or trials involving ≥4 patients presenting with generalized toxocariasis.

Year	Authors & Parameters	Type of the Study	ABZ ^a^	MBZ ^a^	TBZ ^a^	DEC ^a^	NT ^b^	*p*
1987	**Magnaval et al. [84]**	C, Rand ^c^						
	Number of patients			42	30			
	Daily dose			25 mg/kg	25 mg/kg			
	Duration (days)			7	7			
	Minor side effects			10 (23.8%)	18 (60%)			0.004
	Major side effects			none	none			
	Number of cured patients			24 (57%)	15 (50%)			NS ^i^
	Variation in the eosinophil count			NA ^g^	NA ^g^			
1989	**Stürchler et al. [68]**	C, Rand ^c^						
	Number of patients		19		15			
	Daily dose		10 mg/kg		50 mg/kg			
	Duration (days)		5		5			
	Minor side effects		7 (37%)		4 (27%)			NS ^i^
	Major side effects		none		1 (6.7%)			
	Number of cured patients		6 (31.6%)		8 (53.3%)			NS ^i^
	Variation in the eosinophil count		NA ^g^		NA ^g^			
1992	**Magnaval et al. [86]**	C, Rand ^c^						
	Number of patients			45			43	
	Daily dose			10–15 mg/kg			NA ^g^	
	Duration (days)			3 × 6			NA ^g^	
	Minor side effects			26 (57.8%)			13 (30.2%)	0.01
	Major side effects			none			none	
	Variation in the clinical score			−65.6%			−59.2%	NS ^i^
	Variation in the laboratory score			−17.7%			+0.7%	<0.001
1995	**Magnaval [85]**	C, Rand ^c^						
	Number of patients			41		39		
	Daily dose			20–25 mg/kg		3–4 mg/kg		
	Duration (days)			21		21		
	Minor side effects			7 (17.1%)		12 (30.8%)		NS ^i^
	Mazzotti-like reaction ^f^			none		4 (10.25%)		0.05
	Major side effects			none		1 (2.6%)		NS ^i^
	Variation in the clinical score			−68.6%		−70.9%		NS ^i^
	Variation in the eosinophil count			−37.5%		−40.9%		NS ^i^
2003	**Altcheh et al. [69]**	O, Prosp ^d^						
	Number of patients		16 (VLM)					
	Daily dose		15 mg/kg					
	Duration (days)		15					
	Minor side effects		NA ^g^					
	Major side effects		NA ^g^					
	Number of cured patients		16 (100%)					
	Variation in the eosinophil count		−70.4%					
2011	**Turrientes et al. [70]**	O, Retr ^e^						
	Number of patients		4 (VLM)					
	Daily dose		10–15 mg/kg					
	Duration (days)		5					
	Minor side effects		NA ^g^					
	Major side effects		NA ^g^					
	Number of cured patients		4 (100%)					
	Variation in the eosinophil count		−52.7%					
2017	**Kim et al. [73]**	O, Retr ^e^						
	Number of patients		34				11	
	Daily dose		800 mg				NA ^g^	
	Duration (days)		5 to 7				NA ^g^	
	Minor side effects		NA ^g^				NA ^g^	
	Major side effects		NA ^g^				NA ^g^	
	Number of cured patients		NA ^g^				NA ^g^	
	Normalization of the eosinophil count		28 (82.3%)				4 (36.4%)	0.007
2018	**Kroten et al. [71]**	O, Retr ^e^						
	Number of patients		66					
	Daily dose		15 mg/kg					
	Duration (days)		10 × 2					
	Minor side effects		NA ^g^					
	Major side effects		NA ^g^					
	Number of cured patients		47 (71%)					
	Variation in the eosinophil count		NA ^g,h^					
2018	**Yoon et al. [19]**	O, Prosp ^d^						
	Number of patients		141				12	
	Daily dose		800 mg				None	
	Duration (days)		Variable				NA ^g^	
	Minor side effects		NA ^g^				NA ^g^	
	Major side effects		NA ^g^				NA ^g^	
	Number of cured patients		120/141				10/12	NS ^i^
			(85.1%)				(83.3%)	
	Variation in the eosinophil count		NA ^g^				NA ^g^	
2019	**Hombu et al. [72]**	O, Prosp ^e^						
	Number of patients		113 (VLM)					
	Daily dose		15 mg/kg					
	Duration (days)		28 × 2					
	Minor side effects		2 (1.8%)					
	Major side effects		35 (31%)					
	Number of cured patients		91 (80.5%)					
	Variation in the eosinophil count		NA ^g^					
2021	**Phuc et al. [93]**	O, Prosp ^d^						
	Number of patients				80			
	Daily dose				50 mg/kg			
	Duration (days)				5			
	Minor side effects				27(33.8%)			
	Major side effects				none			
	Number of cured patients				63 (78.8%)			
	Variation in the eosinophil count				−41%			

^a^ ABZ; albendazole; MBZ: mebendazole; TBZ: thiabendazole; DEC: diethylcarbamazine—^b^ no treatment (abstention or placebo)—^c^ controlled, randomized—^d^ observational, prospective—^e^ observational, retrospective—^f^ arthralgias, edemas, urticarial—^g^ not applicable or not available—^h^ displayed as graphics—^i^ not significant.

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
