# Peer review of "Therapy and Prevention for Human Toxocariasis"

_microorganisms, 2022, doi:10.3390/microorganisms10020241_

Round 1

Reviewer 1 Report

This article reviews the treatment and prevention of toxocariasis in detail, which has important reference significance. 

However, this review is more like a description than an academic paper. 

This review lacks the latest research results, and the detailed use method of therapeutic drugs does not need to be described in detail, and it is recommended to simplify. 

The diagnosis of toxocariasis should be listed separately. 

Author Response

The authors thank their colleague for the analysis of the manuscript.

Reviewer #1
1. This article reviews the treatment and prevention of toxocariasis in detail, which has important reference significance. However, this review is more like a description than an academic paper. 
We agree this manuscript is not an academic research paper, but an attempt to give the current "state-of-the-art" regarding the therapy of human toxocariasis. Otherwise said, this is a high-grade educational article targeting doctors who are specialized mainly in infectious diseases, internal medicine or tropical medicine.

2. This review lacks the latest research results
Usually, when a reviewer delivers such a remark, she (he) adds one or two examples supporting her (his) claim. This is helpful for the reviewed authors.
Possibly a statement, in the submitted manuscript, escaped reviewer's attention. We wrote in the section "2.2. Efficacy of available anthelmintics for therapy of human toxocariasis" " .... Only relevant studies including four or more cases are discussed below. Anecdotal reports are quoted only if they have a historical relevance or concern NT or OT". Which clearly implies that certain articles from the literature, recent or ancient, were not quoted.
However, we checked again the literature, and we did not find ourselves as having omitted recent publications related to the topic of this manuscript. So we recall the reviewer our manuscript is not a full review of human toxocariasis. 

2.  the detailed use method of therapeutic drugs does not need to be described in detail, and it is recommended to simplify. 
Such as afore-mentioned, this manuscript is intended for providing valuable and hopefully indisputable information to very specialized doctors about the therapeutic options concerning the therapy of toxocariasis. Moreover, such as described in the manuscript, anthelmintic use for treating human toxocariasis is not standardized. Consequently, the future readership certainly is eager to have precise information about anthelmintic drugs that unlike, e.g. antibiotics, are not commonly used. 

3. The diagnosis of toxocariasis should be listed separately. 
Again, this manuscript is not a full review of human toxocariasis. Epidemiology, pathophysiology, clinical and laboratory features and finally the laboratory diagnosis were briefly recalled in the "Introduction" section. Nothing in the topic of the paper would support to have diagnosis described in a distinct section. 

Reviewer 2 Report

Therapy and prevention for human toxocariasis

Magnaval et al. organized knowledges of human toxocariasis, epidemiology, pathophysiology, clinical spectrum, and imaging, or laboratory diagnosis. Human toxocariasis is a worldwide zoonotic helminthiasis caused by infection with Toxocara canis or Toxocara cati larvae. To treat for human toxocariasis, four drugs are registered for human use. In addition, in order to prevent human toxocariasis, parents and children must receive counseling for geophagia, and personal hygiene, including hand washing, is important, especially when handling food or after contacting a dog. It is important to wear appropriate gloves while gardening. In conclusion, they said that the treatment of anthelmintics for human toxocariasis remains somewhat unsatisfactory and that the treatment of anthelmintics for the systemic form of human toxocariasis should be standardized. The reports are well-designed, organized, and descripted. But, several questions were occurred. I have minor questions.

  1. There are minor typing errors in this paper, so please check the paper.
  2. It would be nice to explain why toxocara infection is classified as a systemic or computerized form.
  3. Does VLM by Toxocarasppinfection not appear well in adults even if an infection occurs? Is it because of the immune system?
  4. Methods for diagnosis, such as ELISA, Western blot, and biopsy, will take a long time, but are there any research trends on new diagnostic methods or research to solve this problem?
  5. It would be better if you let me know why in vitro studies focus on larvicidal activity in the first sentence of paragraph 2.1.
  6. I think you said it's effective to eat abz at 220 mg/kg b/w every day, but are you saying that it's effective to actually eat 10-15 mg/kg b/w?
  7. In conclusion, it seems that NT and OT are not often diagnosed, but research on anthelmintic treatment for NT and OT seems to be inappropriate for important reasons.

Author Response

The authors thank their colleague for the analysis of the manuscript.

1. "There are minor typing errors in this paper, so please check the paper".
Following a further Winword check, the ese errors were corrected. Also, perhaps the reviewer pointed out the concomitant use of "helminthic" and "anthelmintic".
"Helminthic" is an adjective indicating the status of an infectious agent or a disease.
"Anthelmintic" is the preferred spelling for the noun or the adjective referring to a drug active on helminths.

2. "It would be nice to explain why toxocara infection is classified as a systemic or computerized form".
We guess your use of "computerized" instead of "compartmentalized", such as written throughout the manuscript,  was a "lapsus calami". Anyway, "compartmentalized" stands for "restricted to some enclosed parts of the body". Such as CNS, that is marked out by the meninges or eye. Consequently, a main feature in the nosography of human toxocariasis is the largest part of patients presenting with ocular or neurological toxocariasis do no have a general involvement, out of these compartments. 

3. "Does VLM by Toxocara spp infection not appear well in adults even if an infection occurs? Is it because of the immune system?"
Likely the reasons of this difference can be found in epidemiology, not in immunopathology. That is, poverty means lack of environmental hygiene resulting in the presence of contaminated dog - or cats - deposits in the soil. Where children play, thus exposing them to repeated and / or large ingestions of Toxocara spp. eggs, sometimes via the "pica" phenomenon. Should you add malnutrition and geophagia to this, the situation worsens.
In contrast, in westernized countries, only few adult people - mostly geophagic mentally deficient - are exposed to the ingestion of such large inoculums resulting in VLM syndrome.

4. "It would be better if you let me know why in vitro studies focus on larvicidal activity in the first sentence of paragraph 2.1"
Such as written in the "Introduction" section, human toxocariasis is due to the presence of Toxocara spp. larvae in the human organism, not of adult worms. Therefore, there is no need for testing the sensitivity of adult Toxocara worms. Only that of larvae is interesting.

5. "I think you said it's effective to eat abz at 220 mg/kg b/w every day, but are you saying tat it's effective to actually eat 10-15 mg/kg b/w?"
There is a great difference between the results from preclinical studies about the efficacy of a given drug, and the conclusions from the following phases of the relevant clinical research, leading to the approval of the compound for human use.
See https://en.wikipedia.org/wiki/Phases_of_clinical_research
for a more detailed information about the assessment of drugs for human use. 

6. "In conclusion, it seems that NT and OT are not often diagnosed, but research on anthelmintic treatment for NT and OT seems to be inappropriate for important reasons".
Exactly. Such as stated in the manuscript at successive levels 1) NT and OT are rare forms of human toxocariasis 2) these forms are mostly severe. Consequently it appears to be impossible, if not very difficult, to carry out prospective studies in this kind of patients. 

Round 2

Reviewer 2 Report

My concerns has been addressed.